# Non-nutritive suck and voice onset time: Examining infant oromotor coordination

**Elizabeth Heller Murray**[1], **Joanna Lewis**[2], **Emily Zimmerman**[2]*

**1** Department of Communication Sciences and Disorders, Temple University, Philadelphia, Pennsylvania, United States of America, **2** Department of Communication Sciences and Disorders, Northeastern University, Boston, Massachusetts, United States of America

* e.zimmerman@northeastern.edu

**Data Availability Statement:** A minimally anonymized data set with subject-level data for voice onset time and non-nutritive suck variables has been made public through the open science

## Abstract

The variability of a child's voice onset time (VOT) decreases during development as they learn to coordinate upper vocal tract and laryngeal articulatory gestures. Yet, little is known about the relationship between VOT and other early motor tasks. The aims of this study were to evaluate the relationship between infant vocalization and another early oromotor task, non-nutritive suck (NNS). Twenty-five full-term infants (11 male, 14 female) completed this study. NNS was measured with a customized pacifier at 3 months to evaluate this early reflex. Measures of mean VOT and variability of VOT (measured via coefficient of variation) were collected from 12-month-old infants using a Language Environmental Analysis device. Variability of VOTs at 12 months was significantly related to NNS measures at 3-months. Increased VOT variability was primarily driven by increased NNS intraburst frequency and increased NNS burst duration. There were no relationships between average VOT or range of VOT and NNS measures. Findings from this pilot study indicate a relationship between NNS measures of intraburst frequency and burst duration and VOT variability. Infants with increased NNS intraburst frequency and NNS burst duration had increased VOT variability, suggesting a relationship between the development of VOT and NNS in the first year of life. Future work is needed to continue to examine the relationship between these early oromotor actions and to evaluate how this may impact later speech development.

## Introduction

The infant suck reflex is one of the earliest motor reflexes to develop, emerging *in utero* around 15 weeks' gestational age [1] and stabilizing around 34 weeks' gestational age [2]. Infants have two types of suck: a nutritive suck used for feeding and a non-nutritive suck (NNS) characterized by the absence of nutrient delivery [1,3]. Infant non-nutritive suck is less complex than nutritive suck as it does not involve swallowing and given that it develops early, provides an early metric into the infants developing oromotor system. NNS is characterized by bursts of suck cycles, occurring at approximately 2 hertz, separated by pause periods for respiration, see Fig 1 [3]. NNS provides a window into central nervous system function, with disordered NNS patterns noted in infants who are preterm or who have neurological impairments [4–7].

framework platform available here: https://osf.io/f2sk7/.

**Funding:** This project was funded by the NIH grants DC016030 (EZ) and EHM's salary was partially supported by DC013017 from the National Institute of Deafness and Other Communication Disorders.

Furthermore, NNS changes throughout the first year of life [8], and these changes are likely due to experience, anatomical growth, and a neurological system that are shifting from reflexive to more cortically driven [9]. In addition to providing information about current neuromotor development, evaluation of infant NNS can also provide information on future functional outcomes. Relationships have been found between infant suck and later oral feeding difficulties [10], language impairment [11–13], intelligence quotient [13], and other cognitive, developmental, and motor delays [14–18]. As infant suck is present at birth, understanding the relationship between the early infant suck measures and later developing speech, language, or cognitive skills will provide valuable information on neurodevelopment in both typical and vulnerable populations.

During this time-period of infant NNS development, changes can also be seen in another motor action that uses overlapping musculature, speech production [19]. Although speech and non-speech tasks have distinct motor activation patterns [20–24], the similarity in the cyclical and rhythmic movements, musculature, and neural processes suggests that understanding the relationship between speech and infant suck can provide valuable information about development. One theoretical model that proposes this relationship is the *Frame Content Theory of the Evolution of Speech Production* [25]. This theory states that the *frame* is the continual mouth open-close rhythmic movement, seen in the jaw and tongue movement in feeding and sucking. As the infant develops and interacts with the environment, the *content* (e.g., vocalizations, verbal output) are superimposed on the *frame* [25]. Thus, this theory suggests early motor action of NNS (*frame*) will be related to the development of speech production (*content*).

Babbling, one of the earliest stages of speech production in which infant produce speech-like oromotor movements, begins around six months of age [26], and typically consists of stop consonant-vowel productions (e.g., /dada/). The timing between the release of the stop consonant and start of the subsequent vowel, called voice onset time (VOT) can be measured in an acoustic signal, providing information on an infant's oromotor control and coordination [27,28]. There are four proposed stages of VOT development, first described by Macken and Barton (1980) and later expanded by Hitchcock and Koenig (2013). During the first stage,

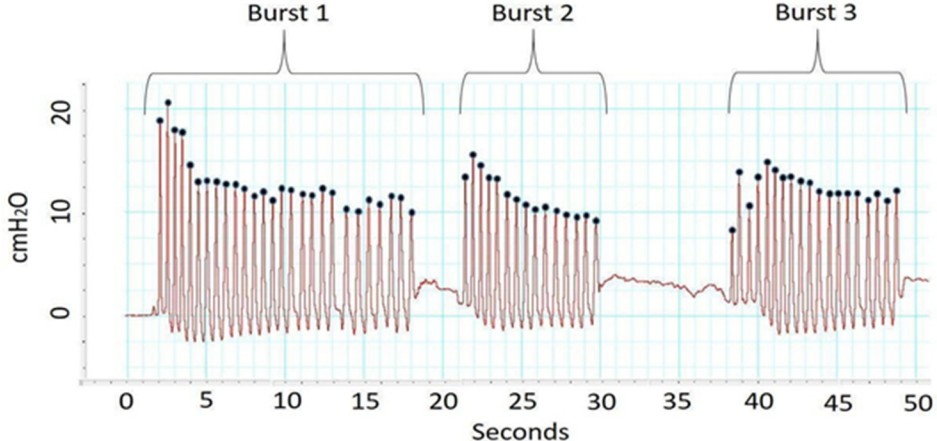

**Fig 1. Example Non-nutritive suck (NNS) bursts: Infant NNS is arranged in bursts of sucking with pause periods for respiration.** Each burst contains cycles within it, which are depicted as black dots in this image. The frequency (Hz) within a burst is measured by the number of cycles per second. The strength of the suck cycle, or amplitude (cmH20), is measured by examining the top of each cycle where the black dot is located. This image depicts 50s of NNS data with 3 NNS bursts. Burst 1 has 27 cycles/burst (burst duration of 16 sec), Burst 2 has 15 cycles/burst (burst duration 8 seconds) and Burst 3 with 19 cycles/burst (burst duration 11 seconds). Amplitude of each cycle (black dot) can be determined looking at the y-axis.

infants' VOTs range from 0 to 20 milliseconds (ms). This stage requires minimal coordination, as the vowel phonation begins almost immediately after the stop consonant closure is released. At this stage, articulatory accuracy of voiced bilabial /b/ and alveolar /d/ productions is more accurate than their unvoiced cognates (/p t/). The second stage involves the beginning of the development of the voicing contrast seen in adults, and infants begin producing voiceless stops with longer VOTs. During the third stage there is the continued elongation of VOTs for voiceless stops. This stage often involves an "overshoot" phase, in which VOTs for voiceless productions are significantly longer than adult productions. The fourth stage emerges around two to three years of age. During this stage, average VOTs are comparable to adults, yet, considerable variability continues to be present until around seven years of age [26,29–32].

The purpose of the current study was to examine if there was a relationship between the early oromotor movements of NNS and babbling, a later developing movement that requires oromotor control. Based on the aforementioned *frame/content* model of speech production evolution [25], we hypothesized that aspects of infant NNS pertaining to cyclical jaw movements, such as NNS cycles/bursts, burst duration and intraburst frequency, at 3-months will be more related to advanced productions VOT productions at 12 months as they are building their *content* on a more mature *frame*. Consistent with previous work examining VOT in children [31], both average and variability metrics will be examined to elucidate information about the infant's VOT developmental stage as well as the variability of their productions. Understanding the relationship between typical development of infant NNS and its relationship to the early motoric gesture of babbling can reveal important information about overall oromotor coordination abilities and provide a more comprehensive basis for understanding future neurodevelopmental outcomes.

## Methods

### Participants

Twenty-five total infants (11 male, 14 female) participated in this study. NNS measurements from eleven of these infants were reported in an earlier paper [8]. Infants were evaluated at 3 months (average age = 3.04 months, range = 2.56–3.76 months) and at 12 months (average age = 11.97 months, range = 11.53–12.33 months). Participants were all born full-term and had an average birthweight of 122.53 ounces (standard deviation (SD) = 18.34 ounces). All infants passed their neonatal auditory screening; by 12 months 57.2% of the infants had a history of ear infections. Hollingshead four-factor index (raw score of 8–66) of socioeconomic status based on marital status, employment status, educational attainment and occupational prestige was on average 57.26 (range = 30–66), and therefore in the mid-to-high SES range. All participants in this study were involved in a larger study examining the relation between early sucking, oral feeding, and vocal development across preterm and full-term infants. Infants were included in the current study if they were: (1) born full-term without congenital or chromosomal anomalies, (2) had usable suck samples at 3 and 12 months, and (3) produced a minimum of ten stop consonants within 20 minutes during their 12-month appointment. This study was approved by the institutional review board at Northeastern University. Participants were recruited by word of mouth, Facebook groups, and flyer distribution. All caregivers provided written consent for the study and were compensated for their participation.

### Data collection

Data collection was completed in the infant's home approximately one hour before a scheduled feed; NNS measurements were collected at the 3-month visit to capture this reflexive motor action, and babbling samples used for VOT analysis were collected at the 12-month

visit. Measurements of the infant's NNS were collected at 3-months with a custom-made research device that consisted of a 0-3-month Soothie pacifier (Philips, Avent) attached to a pressure transducer. The pressure transducer was attached to a data acquisition system (Power Lab, ADInstruments), allowing for real-time visualization of NNS using the LabChart software (ADInstruments). The pressure transducer in the custom-made research device was calibrated with an external pressure calibrator Meriam M1 Series Digital Manometer Calibrator; a range of pressure measurements from the NNS system were recorded simultaneously with both pressure transducers and used to produce a linear calibration curve for the NNS system. Following calibration, parents/caregivers were instructed on how to offer the infant the pacifier, which consisted of demonstration by the research assistant to cradle the infants and offer the infants the pacifier. Researchers encouraged a quiet environment for data collection; however, since the study was completed in the home this was not always possible. Ideally, infants were in a quiet-alert state; however, data collection was discontinued if the infant began to cry, appeared distressed, or rejected the pacifier. Average time infants NNS suck was recorded was 3.09 minutes.

During the 12 month visit, each infant was fitted with a Language Environment Analysis (LENA), a wearable recording device that is widely used in research to analyze early speech vocalizations in young infants [e.g., 33,34]. The LENA device is a small piece of hardware (3-3/8" x 2-3/16" x 1/2") that houses an omnidirectional microphone with a flat 20 hertz (Hz)– 20,000 Hz frequency response and records acoustic data at 16,000 Hz [35]. For each child, the LENA recorder was placed in a dedicated LENA vest; the vest keeps the microphone a consistent distance from the infant and is designed with fabric that has minimal impact on the acoustic recordings [36]. Parents were instructed to leave the vest on their infant for the remainder of the day (with the exception of bath and nap times), continue with their typical routines, and document all activities done while the infant was wearing the vest.

## Measures

**Non-Nutritive Suck (NNS).** Trained experimenters identified NNS burst manually using the LabChart software. We created a study settings file in the LabChart software that consisted of a NNS sample rate of 1000 samples per second with a low-pass filter with a cut off frequency of 50 Hz. NNS physiology has a stereotypical burst-pause pattern, with an intraburst frequency of 2 Hz and each burst containing 6–12 suck cycles [3]. Bursts were defined two or more suck cycles in a row, with the cycles less than one second apart and each cycle's amplitude at least one $cmH_2O$ (see example in Fig 1). This definition of burst is consistent with previous studies examining NNS in young infants [8,37–40]. Following manual selection of all bursts, the best two minutes of NNS data were selected based on cycle number, which is a common procedure used across studies [8,37–39] in an effort to examine the infant's most active NNS sample. NNS measures were calculated with a custom-made NNS burst macro in LabChart. Then, the average of the two minute samples was taken to determine the following NNS minute rates: (1) *burst amount*, the number of NNS bursts in a minute, (2) *burst duration* in seconds (sec), the average length of the burst (3) *cycle amount*, the number of cycles per minute, (4) *cycles/burst*, average number of cycles in each burst per minute, (5) *amplitude ($cmH_2O$)*, average amplitude of the pressure of the cycles, measured as peak-height minus peak-trough in $cmH_2O$, and (6) *Frequency* (Hz), the intraburst frequency between cycles.

**Voice onset time (VOT).** Algorithms in the LENA Pro software were used to identify continuous speech spoken by the infant. The most voluble hour (i.e., the hour with the most infant vocalizations) was found for each infant and the activity log was examined to verify the infant was awake during the selected hour. The most voluble 20 minutes from each infant's

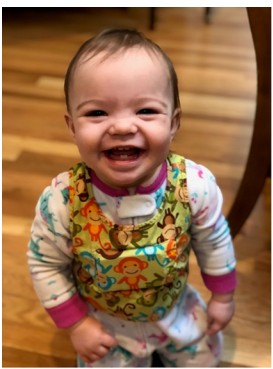
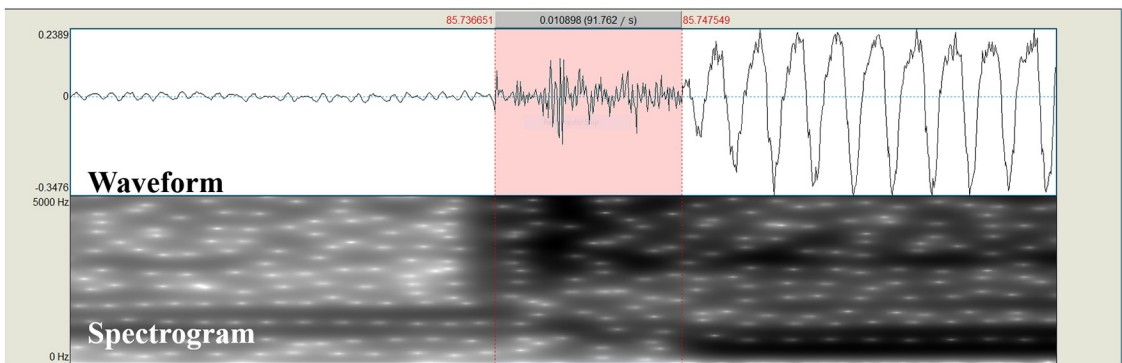

**Fig 2. LENA vest (left) and Voice onset time example (Right).** Left: An infant wearing the LENAvest during recording (The parent of the infant pictured in this manuscript has given written informed consent (as outlined in PLOS consent form) to publish these case details). Right: Praat window with waveform and spectrogram from a stop consonant-vowel production. Voice onset time is measured from the burst to the start of the subsequent vowel, indicated by the highlighted portion.

most voluble hour was exported to Praat [41] for acoustic analysis. An initial rater identified all stop consonants from the acoustic waveforms and spectrograms. Default spectrogram Praat settings were used, with a view range of 0–5000 Hz, dynamic range of 70 dB, and a window length of 0.005 seconds. VOT, defined as the time between the burst of the stop consonant and the start of the subsequent vowel, was measured for each stop consonant-vowel pair identified (Fig 2). To control for differences in the amount of stop consonant-vowel productions between infants, only the first ten stop consonant-vowel productions were selected for evaluation. Each stop consonant selected was identified based on both the rater's auditory-perception of the production and the presence/absence of features in the spectrogram (e.g., voicing bar). The majority of productions were perceived as a voiced /d/ production, with voiced phonemes identified more frequently than voiceless consonants (Table 1). Stop consonants were identified from segments of reduplicated (e.g., baba) and variegated (e.g., baga) babbling. No identifiable words were noted with stop consonants. A second rater (one of the senior authors) reviewed all VOTs and adjusted when needed. Each rater repeated analysis on 20% of the participants, for a total of 50 VOTs repeated. The average absolute difference in VOT ratings were calculated to assess intra-rater reliability (*rater 1*: average VOT difference = 0.97 ms, standard deviation = 2.0 ms; *rater 2*: average VOT difference = 0.91 ms, standard deviation = 1.3 ms).

Three measures were used to evaluate VOT productions. Average VOT and VOT range (maximum VOT–minimum VOT) were calculated to provide information on the infant's stage of VOT development. Longer VOTs and larger VOT ranges were interpreted as

**Table 1. Perception of stop consonants during initial calculation of voice onset time.**

|  | Phoneme | Phoneme Count | Infant Count |
|---|---|---|---|
| Voiced | b | 46 | 15 |
|  | d | 138 | 23 |
|  | g | 26 | 6 |
| *Total Voiced* |  | 210 | 25 |
| Voiceless | p | 2 | 2 |
|  | t | 32 | 16 |
|  | k | 2 | 2 |
| *Total Voiceless* |  | 36 | 16 |
| Ambiguous | d/t | 4 | 2 |
| *Total (Voiced and Voiceless)* |  | 250 | 25 |

advanced development as the infants were prolonging VOT productions. To evaluate the variability of the infants' productions, the coefficient of variation (CoV) of VOT was calculated for each infant. CoV, that is the standard deviation VOT divided by the mean VOT, providing a metric of infant variability of productions *around* their mean production. Therefore, using CoV allows for evaluation of infant VOT variability, while adjusting for individual differences in average VOT.

### Data analyses

Descriptive statistics on NNS measures at 3 months and VOT measures at 12 months were completed. All measures were converted to rank order for the subsequent analyses to account for the lack of normality in the measures. Correlations examined the relationships among NNS measures; a Bonferroni corrected alpha level of 0.0033 (0.05/15 correlations = 0.0033) was used to correct for multiple comparisons. Three multiple linear regressions examined whether NNS measures at 3 months predicted either average VOT, range of VOT, or CoV of VOT at 12 months. All analyses were completed in JMP Pro [42].

### Results

#### Analysis of individual measures

Average VOT at 12 months was 7.82 ms across all infants, with individual infant averages ranging from 0 ms to 33.87 ms. Examination of individual productions indicated infants produced a large range of VOT values (Fig 3), with average CoV of VOT measured at 1.22.

Infants at 3 months produced NNS an average of 4.10 bursts (range: 1.50–9.50) per minute, with an intraburst frequency average of 2.06 Hz (range: 1.36–2.75), average burst duration of 4.93 seconds (range: .94–11.97). Infants produced an average of 10.02 cycles per burst (range: 2.25–27.17), 42.24 cycles per minute (range: 3.5–109.50), and an average cycle amplitude of 12.32 $cmH_2O$ (range: 1.19–28.03).

#### Relationships between NNS measures at 3 months and VOT measures at 12 months

The NNS measure of burst duration was highly correlated with NNS measures of cycle amount ($r = .84$, $p < 0.001$) and cycles/burst ($r = .97$, $p < 0.001$). The NNS measure of cycles amount was highly correlated with burst amount ($r = .71$, $p < 0.001$) and cycles/burst ($r = .87$, $p < 0.001$, Table 2). Based on the high association between cycle amount and cycles/burst with other NNS measures, only the NNS measures of burst duration, frequency, amplitude, and burst amount were included in the regression model. There was no significant effect of NNS measures of burst duration, frequency, amplitude, burst amount on either average VOT or range of VOT (all $p > 0.05$). A regression model including the NNS measures of burst duration, frequency, amplitude, and burst amount significantly predicted CoV of VOT ($F(4,18) = 3.613$, $p = 0.02$), with an $R^2 = 0.45$ (Fig 4). Increased variability of VOT was driven by increased NNS burst duration ($\beta = 0.53$, $p = 0.008$) and increased NNS intraburst frequency ($\beta = 0.50$, $p = 0.01$). Measures of NNS height ($\beta = 0.21$, $p = 0.32$) and decreased NNS burst amount ($\beta = -0.34$, $p = 0.10$) did not reach significance in this model.

### Discussion

This study provided a novel look into the relationships between two early oromotor actions, NNS and babbling. Findings from the current study indicated a relationship between NNS at 3-months and VOT variability at 12-months. The metric of variability used in the current

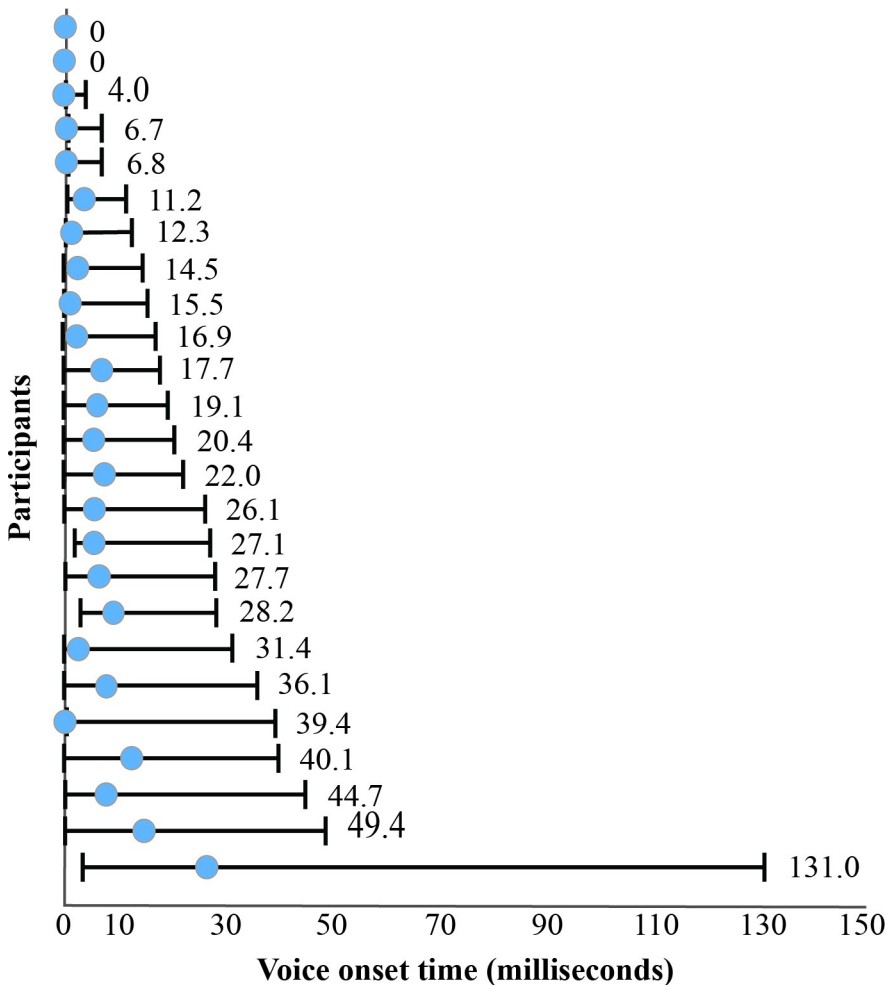

**Fig 3. Infant voice onset times.** Voice onset time medians (blue dots) and ranges for each infant.

work was coefficient of variation (CoV), a measure that examines variability while controlling for average VOT. Previous work examining VOT productions suggests that VOT variability decreases during maturation [26,43,44]; however, this reduction in VOT variability occurs after two years of age [45]. Although the decrease in variability in older children has been associated with improved accuracy of productions [44,46,47], it is unlikely that differences in

**Table 2. Correlation matrix of non-nutritive suck (NNS) measures.**

|  | Burst Duration (sec) | Frequency (Hz) | Amplitude (CmH$_2$0) | Burst Amount | Cycles/Burst |
|---|---|---|---|---|---|
| Frequency (Hz) | -.13 | – | – | – | – |
| Amplitude (CmH$_2$0) | .18 | -.26 | – | – | – |
| Burst Amount | .37 | .001 | .47 | – | – |
| Cycles/Burst | **.97**[*] | .06 | .14 | .38 | – |
| Cycle amount | **.84**[*] | .09 | .34 | **.71**[*] | **.87**[*] |

[*] correlations significant at p < 0.0033.

Bolding indicates significant correlations.

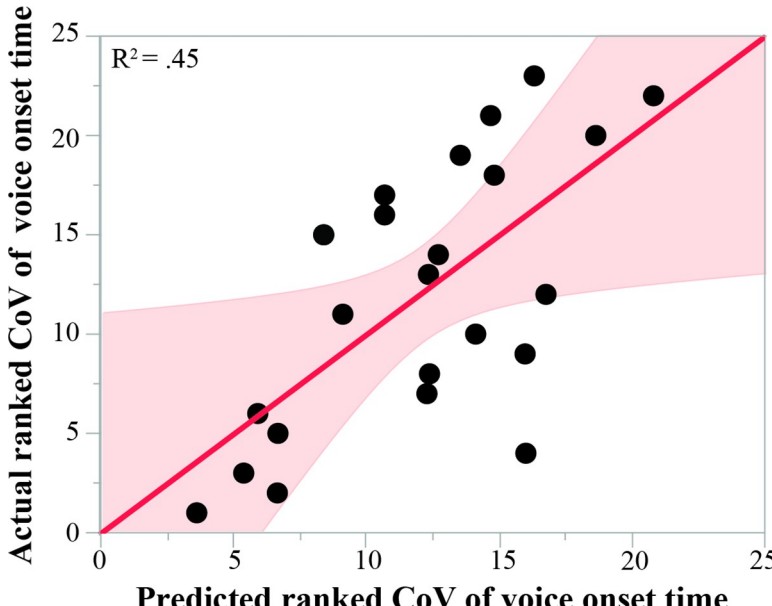

**Fig 4. Relationships between voice onset time variability and non-nutritive suck.** Relationship between the actual ranked coefficient of variation (CoV) of voice onset time and the CoV of voice onset time predicted by the regression model that included NNS measures of burst duration, frequency, amplitude, and burst amount. NNS measures of burst duration and intraburst frequency were significant predictors. Shaded area represents 95% confidence interval around the regression line.

variability of babbling in 12-month-old infants is due to refining movements in order to accurately reach a target. Instead, we suggest that the increased variability evident during this period may be due to increased exploration while infants are acquiring new articulatory movements. This interpretation is based on *Dynamic Systems Theory*, which posits that learning a new skill is preceded by increased variability of movement [48,49]. Therefore, we speculate that increased VOT variability in the current study may be indicative of a more advanced stage of speech production acquisition.

Findings from this study indicated that infants with increased VOT variability at 12 months had increased NNS at 3 months, driven by increased NNS burst duration and increased NNS intraburst frequency. Put simply, infants who produced longer bursts, which likely consisted of more NNS cycles per burst, produced at a faster rate had increased VOT variability during babbling. Both NNS and babbling involve jaw and tongue movements; the jaw and tongue movements required to produce multiple, rapid cycles within a burst mirrors the jaw and tongue movements during babbling [20,50–52]. Interpreted within the *frame/content* theory of evolution of speech production [25], we postulate that ability to produce longer bursts at a faster rate at 3-months may allow the infant's *frame* to be primed for the development of the *content* of babbling at 12-months. That is, infants are building this later *content* (babbling) on the earlier *frame* (NNS). However, further work is needed to examine this relationship beyond the scope of this pilot study. The current work used the measure of VOT, which allows inference about jaw movement, yet this temporal measure does not provide significant information about tongue control. As multiple orofacial structures are involved in both infant sucking and babbling (e.g., jaw, lip, tongue), future work measuring the kinematic movement of all structures involved in both NNS and speech production is needed to further evaluate how these early motor actions are related.

Although relationships were found between NNS and CoV of VOT, the current study did not find any relationship between VOT average or range of VOT productions and NNS measures. Based on the proposed stages of VOT development [30,32], an average VOT value of 7.82 ms suggests infants are in stage one of VOT development, in which the vowel phonation begins almost immediately after the stop consonant closure is released. One possible reason we did not find a relationship between NNS measures and VOT averages is that the small differences in VOT averages at stage one may not provide a meaningful metric of development. Thus, to evaluate the relationship between VOT development further, we examined whether VOT range would relate to NNS measures. Consistent with previous studies, individual infant averages ranging from 0 ms to 33.87 ms [32,45,53–56] indicating that infants were beginning to lengthen the time between stop closure offset and vowel onset [30,32]. However, VOT range did not meaningfully relate to NNS productions. A potential reason for the absence of findings may be the lack of a clear target for babbling, thus making interpretation of VOT range more complex. Previous work examining VOT in the first few years of life has mainly focused on production of words [32,45,53,55], with the few articles that discussed VOTs of babbling focusing primarily on cross-linguistic differences [54,56]. The evaluation of VOT in words allows for judgements on accuracy of productions as the intended target is known, whereas babbling does not have to have a clear target. Therefore, future work is needed to longitudinally examine the relationship between the NNS reflex and more intentional early speech production produced at later time-points and across patient populations.

It should be noted that although we have discussed our results within the *frame/content* theory, there are other potential explanations that require consideration. First, it is possible that the measurements of NNS and VOTs are capturing development of a single skill, rather than the building of the *content* (babbling) on the *frame* (NNS). Further work is needed to determine if the skill of babbling is built on the skill of NNS, or if these are purely two types of oromotor movement measurements captured at different points in time. Second, it is important to acknowledge that in focusing on the oromotor relationships between NNS and VOT productions, the current work does not address the impact of vocal fold movement on VOTs. As VOT depends on the coordination of the articulatory and vocal fold movement, some of the findings of the current work may be related to vocal fold changes. During the first few years, the vocal folds are undergoing significant structural changes that may impact their flexibility and movement [57,58]. Thus, further work is needed to clarify the potential impact of vocal fold development on VOTs during babbling. This future work should also include an examination of vowel token as emerging work suggests that, unlike adults [59–61], the relationship between VOTs and vowels in children is not clear [62]. As these differences in vowels may be related to intrinsic fundamental frequency differences, as well as differences in vocal tract positioning (and subsequent formant measurements), understanding this relationship may provide valuable information about VOT development. Lastly, although the consonants were labeled during the initial identification of VOT instances, they were not part of a larger perceptual study and therefore we considered them as preliminary labels for the consonants. Due to the absence of a clear consonantal target and the sparsity of different consonants identified during preliminary labeling, we did not pursue additional analysis of any potential relationships between specific consonants and NNS measures. As there is evidence that VOT may vary by place [e.g., 32,62–66], future work that includes methodology to confirm place of articulation (e.g., electromagnetic articulography, video analysis) is needed to examine any potential relationship between place of articulation and NNS measures.

Overall, findings from this pilot study suggests the relationship between NNS and VOT in babbling and/or early words needs further exploration. Understanding this relationship could provide valuable information on the development of the motor control system as a whole and

provide a marker for children who may be at higher risk for later difficulty with speech motor control. Previous work examining VOT in older children has shown differences in VOT control. For instance, evaluation of VOTs in children at high-risk for developing autism spectrum disorder had deceased distinction in voice and voiceless productions [67]. In addition, children who were later diagnosed with childhood apraxia of speech were found to use less voiceless sounds in early productions [68]. Future work should evaluate whether these later appearing differences in VOTs are related to variability in infant babbling, and thereby related to NNS outcomes sampled soon after birth. In this way, NNS and early VOT metrics could potentially serve as early biomarkers for subsequent speech development.

## Limitations

Limitations in the study included a small sample size, as only 25 total infants were included in the study. Due to this small sample size, this study did not examine potential differences in other variables such as infant sex, feeding method, or birthweight on NNS and VOT productions. Further work on a larger sample size is needed to examine whether other relevant variables impact the relationship between early oromotor actions. The current study evaluated NNS measures at 3-months and babbling productions at 12-months and thus, if the infant had a difficult day (e.g., tired or fussy), their data may not be truly representative of their NNS and VOT productions. Future work should also assess both NNS and VOT measures at multiple time points to more comprehensively measure these early motor actions. Finally, infants were recruited from the Northeast through flyers, online parent communities, and word of mouth. This recruitment strategy resulted in only middle to high SES participants; future work is needed with infants from a broader range of backgrounds to increase the generalizability of these findings. Lastly, this study was completed in the home environment. Future work is needed to examine babbling in a more controlled environment, allowing for examination of other factors that may influence babbling measurements (e.g., rate of speech, use of infant directed speech, number of people present in the room). While use of the LENA system in the home provides a window into the infant's natural environment, future work can examine situations where the babbling productions are elicited and more closely controlled.

## Conclusion

The results of this pilot study reveal that there is a relationship between VOT and NNS measures in infants. Increased variability of VOT productions at 12 months was related to NNS measures, with the relationship driven by increased NNS intraburst frequency and increased NNS burst duration at 3-months. No relationships were found between the average or range of VOT productions at 12 months and NNS measures at 3 months. Findings suggest a link between infant vocal development and oromotor movements evident in NNS productions, motivating the need for future work to continue to examine this relationship.

## Acknowledgments

The research team would like to thank the infants and their families for participating in this study. Thanks to Andie Chao for her help on this manuscript. We would also like to thank members of the Speech and Neurodevelopment Lab at Northeastern University for assisting with data collection and analyses.

## Author Contributions

**Conceptualization:** Elizabeth Heller Murray, Emily Zimmerman.

**Data curation:** Emily Zimmerman.

**Formal analysis:** Elizabeth Heller Murray, Joanna Lewis.

**Funding acquisition:** Elizabeth Heller Murray, Emily Zimmerman.

**Investigation:** Joanna Lewis, Emily Zimmerman.

**Methodology:** Elizabeth Heller Murray, Joanna Lewis, Emily Zimmerman.

**Project administration:** Emily Zimmerman.

**Resources:** Emily Zimmerman.

**Software:** Emily Zimmerman.

**Supervision:** Elizabeth Heller Murray, Emily Zimmerman.

**Validation:** Elizabeth Heller Murray, Emily Zimmerman.

**Writing – original draft:** Elizabeth Heller Murray, Joanna Lewis.

**Writing – review & editing:** Elizabeth Heller Murray, Emily Zimmerman.

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
