## [Decision Letter · Decision Letter 0]

22 Mar 2021

PONE-D-21-02855

Non-nutritive suck and voice onset time: Examining infant oromotor coordination

PLOS ONE

Dear Dr. Zimmerman,

Thank you for submitting your manuscript to PLOS ONE. After careful consideration, we feel that it has merit but does not fully meet PLOS ONE’s publication criteria as it currently stands. Therefore, we invite you to submit a revised version of the manuscript that addresses the points raised during the review process.

We look forward to receiving your revised manuscript.

Kind regards,

Michael Döllinger, Ph.D.

Academic Editor

PLOS ONE

Journal Requirements:

3.  We note that Figure 2 includes an image of a [patient / participant / in the study]. 

Additional Editor Comments (if provided):

Reviewers' comments:

Reviewer's Responses to Questions

**Comments to the Author**

1. Is the manuscript technically sound, and do the data support the conclusions?

Reviewer #1: Yes

Reviewer #2: Yes

2. Has the statistical analysis been performed appropriately and rigorously? 

Reviewer #1: Yes

Reviewer #2: Yes

3. Have the authors made all data underlying the findings in their manuscript fully available?

Reviewer #1: No

Reviewer #2: Yes

4. Is the manuscript presented in an intelligible fashion and written in standard English?

Reviewer #1: Yes

Reviewer #2: Yes

5. Review Comments to the Author

Reviewer #1: Nicely designed and completed study that is clearly described. Please consider these few minor points:

1) It would help readers to depict the various measures as part of Fig 1 (or in addition to fig 1).

2) On line 205, the CoV is assigned units (ms). Isn't it a ratio and unit-less?

3) Please describe the consonants - are these consonant-like sounds or would they perceived as actual consonants? Do they generally sound voiced or a mix of voiced and voiceless? Are they primarily reduplicated babbled syllable trains?

4) Discussion - lines 257-260: it sounds as if you're suggesting this is causal (with the sucking, they are building the frame needed for increased babbling). You might want to acknowledge that they might both be consequences of the same underlying process / skill / advantage.

5) I didn't see the NNS individual data at the level I think PLOS requests. Perhaps it is in an appendix that I missed.

Reviewer #2: Thankyou for giving me the opportunity to review this very well written manuscript. It was a pleasure to review.

The manuscript presents the results of a very challenging study comparing the activity of non nutritive sucking in infants at 3 months and the variability of voice onset time in babbling in the same infants at age 12 months.

The authors are to be congratulated for achieving difficult data collection.

All aspects of the study appear to have been well conducted with good scientific rigour.

I have only a few comments that I think the authors should consider.

The stop consonants (if determinable) that VOT was calculated on is not reported. Given that VOT varies across phonemes, it is important that these be identified and categorised if possible and then VOT values in each phoneme group be compared to child values reported in the literature to assess if the infants variability is consistent with phoneme variability ranges. I would be interested to see if any of the NNS measures were more or less correlated with different phonemes. If this is not possible, the authors should provided more detail regarding whether a phoneme target is identifiable or not.

Interpretation of the results seem appropriate regarding the movement of jaw and tongue between NNS and VOT in babbling.

The authors fail, however, to discuss movement of the vocal folds, and the relationship or not of vocal fold movement to jaw and tongue movement. This is a problematic as VOT is a measure of co-ordination of movements of jaw and tongue vocal fold closure with closure and vibration of the vocal folds. They should mention how the onset of phonation may be affected by the articulatory movements common to NNS and VOT. Some comment on vocal fold activity or posture in NNS should also be mentioned for context.

Thankyou again for the opportunity to review this manuscript.

6. PLOS authors have the option to publish the peer review history of their article (what does this mean?). If published, this will include your full peer review and any attached files.

Reviewer #1: No

Reviewer #2: No

---

## [Author Response · Author response to Decision Letter 0]

27 Mar 2021

We would like to thank both the reviewers and the editor for this review of our article Non-Nutritive Suck and Voice Onset Time: Examining Infant Oromotor Coordination. We have addressed each point below and have highlighted the changes in the ‘Revised Manuscript with Track Changes’ document. We are grateful for the time and effort put into these reviews and believe that these changes make the manuscript stronger. Thank you for the opportunity to resubmit. 

1. A minimally anonymized data set with subject-level data for voice onset time and non-nutritive suck variables has been made public through the open science framework platform available here: https://osf.io/f2sk7/

2. We have updated the methods to explicitly say “The parent of the infant pictured in this manuscript has given written informed consent (as outlined in PLOS consent form) to publish these case details.” We have added this information in our ethics statement and have saved a copy of this form securely in the individual’s case notes. 

3. In re-checking all of our analyses prior to uploading the final anonymized dataset, we realized we had reported numbers that were not the final analysis. These resulted in very minor changes to the variables reported in the text and figure 4. These minor corrections did not change the outcomes of the statistical tests. We have highlighted these changes in the manuscript and apologize for this oversight.

Below are the responses to the individual reviewer comments. 

Reviewer #1: 

1. Nicely designed and completed study that is clearly described. Please consider these few minor points

Response: Thank you for your review.

2. It would help readers to depict the various measures as part of Fig 1 (or in addition to fig 1).

Response: We have added additional text to the figure caption to explain these measures in more detail.

3. On line 205, the CoV is assigned units (ms). Isn't it a ratio and unit-less?

Response: Thank you for noticing this oversight. You are correct CoV is unitless and we have updated the text.

4. Please describe the consonants - are these consonant-like sounds or would they perceived as actual consonants? Do they generally sound voiced or a mix of voiced and voiceless? Are they primarily reduplicated babbled syllable trains?

Response: Thank you for this excellent point. We have added additional text in the methods and discussion to address this comment. The consonants were analyzed were from single syllable productions, or instances of reduplicated or variegated babbling. No identifiable words with stop consonants were found in these recordings. We have also added a table detailing the consonants preliminarily identified during the initial VOT analysis.

5. Discussion - lines 257-260: it sounds as if you're suggesting this is causal (with the sucking, they are building the frame needed for increased babbling). You might want to acknowledge that they might both be consequences of the same underlying process / skill / advantage.

Response: Thank you for this note – we agree this important to acknowledge. We have updated the discussion to include an acknowledgement that these could be the same process developing.

6. I didn't see the NNS individual data at the level I think PLOS requests. Perhaps it is in an appendix that I missed.

Response: Thank you for pointing this out. We have added the individual data to an ‘open science framework’ data repository https://osf.io/f2sk7/

Reviewer #2: 

1. Thank you for giving me the opportunity to review this very well written manuscript. It was a pleasure to review. The manuscript presents the results of a very challenging study comparing the activity of non nutritive sucking in infants at 3 months and the variability of voice onset time in babbling in the same infants at age 12 months. The authors are to be congratulated for achieving difficult data collection. 

Response: Thank you for your review.

2. The stop consonants (if determinable) that VOT was calculated on is not reported. Given that VOT varies across phonemes, it is important that these be identified and categorized if possible and then VOT values in each phoneme group be compared to child values reported in the literature to assess if the infants variability is consistent with phoneme variability ranges. I would be interested to see if any of the NNS measures were more or less correlated with different phonemes. If this is not possible, the authors should provided more detail regarding whether a phoneme target is identifiable or not

Response: Thank you for this excellent point. We have also added a table detailing the consonants preliminarily identified during the initial VOT analysis and added text in the discussion. Although we agree that analyzing the relationship between NNS measures is a very interesting idea, we are hesitant to do this in the current paper due to the sample size and the subjective nature of consonant productions in babbling. We have added text to addressed this directly in the discussion and to highlight the importance of doing this in future work.

3. Interpretation of the results seem appropriate regarding the movement of jaw and tongue between NNS and VOT in babbling. The authors fail, however, to discuss movement of the vocal folds, and the relationship or not of vocal fold movement to jaw and tongue movement. This is a problematic as VOT is a measure of co-ordination of movements of jaw and tongue vocal fold closure with closure and vibration of the vocal folds. They should mention how the onset of phonation may be affected by the articulatory movements common to NNS and VOT. Some comment on vocal fold activity or posture in NNS should also be mentioned for context.

Response: Thank you for this comment. We agree that not speaking about vocal fold motion was an oversight and we have now added additional text in the discussion.

---

## [Editor Report · Decision Letter 1]

8 Apr 2021

Non-nutritive suck and voice onset time: Examining infant oromotor coordination

PONE-D-21-02855R1

Dear Dr. Zimmerman,

We’re pleased to inform you that your manuscript has been judged scientifically suitable for publication and will be formally accepted for publication once it meets all outstanding technical requirements.

Kind regards,

Michael Döllinger, Ph.D.

Academic Editor

PLOS ONE

Additional Editor Comments (optional):

Dear authors,

I recommend acceptance, However, please check on following rather small errors:

line 224: "1.190" or "11.90"; if "1.190" is correct, please change to "1.19"

Line 389: please provide pages

line458: "2012;2012" twice? correct?

line472: please provide pages

reference 50: If this is a book, than there is information missing, i guess.

Figure 3: units of x-axis are missin, I guess it is ms

Figure4: units for x-y-axes?

---

## [Editor Report · Acceptance letter]

13 Apr 2021

PONE-D-21-02855R1 

Non-nutritive suck and voice onset time: Examining infant oromotor coordination 

Dear Dr. Zimmerman:

I'm pleased to inform you that your manuscript has been deemed suitable for publication in PLOS ONE. Congratulations! Your manuscript is now with our production department. 

Kind regards, 

on behalf of

Dr. Michael Döllinger 

Academic Editor

PLOS ONE